# Association between Resting Heart Rate and Colorectal Cancer: Results from a Case-Controlled Study

**DOI:** 10.3390/ijerph16162883

**Published:** 2019-08-12

**Authors:** Yu-Jin Kwon, Hye Sun Lee, Mi Ra Cho, Si Nae Kim, Justin Y. Jeon, Nam Kyu Kim, Ji-Won Lee

**Affiliations:** 1Department of Family Medicine, Yonsei University College of Medicine, Seoul 03722, Korea; 2Department of Medicine, Graduate School of Yonsei University College of Medicine, Seoul 03722, Korea; 3Department of Family Medicine, Yong-In Severance Hospital, Gyeonggi 17046, Korea; 4Biostatistics Collaboration Unit, Department of Research Affairs, Yonsei University College of Medicine, 50-1 Yonsei-ro, Seodaemoon-gu, Seoul 03722, Korea; 5Department of Sport Industry Studies, Yonsei University, Seoul 03722 Korea; 6Division of Colorectal Surgery, Department of Surgery, Severance Hospital, Yonsei University College of Medicine, Seoul 03722, Korea

**Keywords:** colorectal cancer, colon cancer, rectal cancer, resting heart rate

## Abstract

Previous studies evaluating associations between resting heart rate (RHR) and cancer-related mortality/prognosis have yielded conflicting results. We investigated whether elevations in RHR are associated with colorectal cancer (CRC). We conducted a case-controlled study involving 1241 CRC patients and 5909 cancer-free controls from the Korean National Health and Nutrition Examination Survey. After propensity score (PS) matching, 1207 CRC patients and 1207 matched controls were analyzed. Associations between RHR and CRC, colon, and rectal cancer were analyzed in appropriate patient subgroups using multiple and conditional logistic regression. Receiver operating characteristics analysis yielded the optimal RHR cut-point to predict CRC. RHR was significantly higher in CRC, colon, and rectal cancer patients than in controls (72.7 bpm in CRC, 72.8 bpm in colon cancer, 72.3 bpm in rectal cancer, and 68.7 bpm in controls; all *p* < 0.001). Analysis of data prior to PS matching yielded the following odds ratios (ORs) per RHR increment for CRC, colon, and rectal cancer: 1.043 (95% confidence intervals (CIs): 1.036–1.049), 1.045 (95% CI: 1.037–1.053), and 1.040 (95% CI: 1.030–1.051), respectively, in unadjusted models, and 1.043 (95% CI: 1.034–1.051), 1.046 (95% CI: 1.037–1.055), and 1.040 (95% CI: 1.027–1.052), respectively, in multivariable adjusted models. Patients with CRC, colon, and rectal cancer have a significantly higher RHR compared to cancer-free controls.

## 1. Introduction

Colorectal cancer (CRC) is the third most-common cancer and second leading cause of cancer-related death in both sexes according to Global cancer statistics GLOCAN 2018 reports [1]. CRC is three times more prevalent in developed countries than in developing countries [1]. Although the exact etiology of CRC remains unknown, CRC development is closely associated with not only dietary habits and physical activity, but also chronic inflammation and insulin resistance [2,3]. High resting heart rate (RHR) may reflect an underlying autonomic dysfunction and sympathetic overactivity [4]. This autonomic imbalance may contribute to increased subclinical inflammation, which has been known to be a central process in the initiation and progression of atherosclerosis [5,6,7,8,9,10]. High RHR increases the risk of cardiovascular diseases (CVD) by elevating arterial wall stress and disrupting vulnerable plaque [11,12]. Many studies have demonstrated the role of RHR for CVD and related mortality [5,6,7,8,9]. Several studies have suggested that high RHR is associated with not only CVD-related death, but also death from other causes, especially cancer mortality [13,14,15,16]. Activated sympathetic nervous system closely linked to elevated heart rate could contribute to the initiation and progression of cancer [17]. Several studies have also suggested a positive role for beta-blockers in reducing risk of cancer development/progression [18,19].

However, although studies have suggested links between high RHR and CRC risk, there are conflicting reports regarding the association between RHR and CRC [20,21,22,23]. A German population-based study showed no significant benefit of beta-blocker use at diagnosis for CRC prognosis among all CRC patients; however, this study suggested a slight benefit of beta-blocker use for prognosis in stage IV CRC patients [24]. Although a recent prospective study also failed to show a significant association between high RHR and cancer-related mortality [20], the study suggested that high RHR may be associated with higher CRC incidence (hazard ratio (HR) for CRC incidence per 10 beats per minute (bpm) increase in RHR was 1.15; 95% confidence interval (CI): 0.97–1.36). A Korean study of CRC survivors showed that elevated RHR was independently associated with a higher rate of advanced adenoma recurrence during a mean follow-up period of eight years [25].

However, to the best of our knowledge, no study to date has compared RHR in CRC patients before any treatment (e.g., surgery) with that of cancer-free controls. The current study investigates RHR differences between patients with confirmed CRC and cancer-free controls participating in the Korean National Health and Nutrition Examination Survey (KNHANES). We further conducted propensity score (PS) matching (for age and gender) to confirm this association in a PS-matched sub-cohort.

## 2. Materials and Methods

### 2.1. Study Population and Data Collection

A total of 1241 CRC patients (712 men, 529 women) who visited the Department of Colorectal Surgery of Severance Hospital, Seoul, between January 2013 and December 2014 were included in the final analysis. A total of 5960 controls were selected from the 2013–2014 KNHANES conducted by Korea Centers for Disease Control and Prevention (KCDC) [26,27]. Patients with the following conditions were also excluded: (a) A history of malignancy, (b) CVD (e.g., stroke, angina, myocardial infarction), (c) other diseases that may influence RHR (e.g., thyroid diseases), (d) body mass index (BMI) < 15 kg/m^2^, (e) arrhythmia history, RHR > 100 bpm, and <40bpm, and (f) systolic blood pressure (SBP) ≥ 180 mmHg or diastolic blood pressure (DBP) ≥ 120 mmHg.

### 2.2. Ethical Approval

This study was approved by the institutional review board of Severance Hospital, Yonsei University (Seoul, Korea) and informed consent was obtained from all participants (IRB No: 4-2016-0859).

### 2.3. Definition of CRC

Colon cancer was defined as tumor in the cecum, appendix, ascending colon, hepatic flexure, transverse colon, splenic flexure, descending, and sigmoid colon. Rectal cancer was defined as tumors occurring at the rectosigmoid junction or rectum, and the total cohort of patients with colon or rectal cancer were defined as CRC patients. Patients with anal canal cancer were excluded. Colon and rectal cancer staging were classified from 0 to III based on the seventh edition of the American Joint Committee on Cancer staging system (AJCC-7) [28].

### 2.4. Covariates

SBP and DBP were measured in a sitting position after resting for at least 5–10 min. In KNHANES, RHR was recorded in the right radial for 15 s and multiplied by 4 to yield RHR in bpm. RHR was measured for 60 s when the RHR was irregular (e.g., less than 15 times for 15 s or more than 26 times for 15 s). In the CRC cohort, the patients’ RHR was measured for 60 s by physicians after sufficient rest (more than 10 min). All physical measurement data were obtained in the clinics during the patients’ first visit before surgery or any other treatment. BMI was calculated as body weight divided by square of height (kg/m^2^). Weight status was categorized as follows based on the BMI criteria for Asians of the Regional Office for the Western Pacific Region of World Health Organization [29]: Underweight (<18.5 kg/m^2^), normal (18.5–23 kg/m^2^), overweight (23–25 kg/m^2^), and obese (>25 kg/m^2^). Blood tests were performed after 8-hour fasting. Alcohol and smoking status were obtained from a self-reported questionnaire. Alcohol consumption status was classified as never-drinker, ex-drinker, and current drinker. Smoking status was categorized as never-smoker, ex-smoker, and current smoker. Hypertension (HTN) was defined as SBP ≥ 140 mmHg or DBP ≥ 90 mmHg or use of hypertension medication or “diagnosed by physician”. Diabetes was defined as fasting glucose ≥ 125 mg/dL or use of diabetes medication or “diagnosed by physician”. Dyslipidemia was defined as total cholesterol (TC) ≥ 240 mg/dL or use of antilipidemic medication or “diagnosed by physician”.

### 2.5. Statistical Analysis

All statistical analyses were performed using SAS software version 9.3 (SAS Institute Inc., Cary, NC) or R package version 3.2.2 (http://www.R-project.org). Data are presented as mean ± standard deviation (SD) or number (%). Clinical characteristics were compared with independent two sample t-test for continuous variables or chi-square test for categorical variables in the PS-unmatched data. We performed 1:1 PS matching to mitigate the confounding effects of age and gender using a nearest-neighbor matching algorithm. The matched demographic characteristics were compared using the paired t-test for the continuous variables or McNemar’s test for categorical variables. Differences in mean value of RHR were calculated using a linear mixed model with covariance pattern (unstructured) after adjusting for age, gender, weight status, alcohol, smoking, HTN, diabetes, and dyslipidemia in the PS-matched dataset. The odd ratios (ORs) and 95% CIs for CRC, colon, and rectal cancer per bpm increment of RHR were calculated by multiple logistic regression model in PS-unmatched data and by conditional multiple logistic regression models in our PS-matched dataset after adjusting for age, gender, weight status, alcohol, smoking, HTN, diabetes, and dyslipidemia. Additionally, we performed receiver operating characteristics (ROC) analysis to determine the optimal cut-off value of RHR that predicts CRC, colon, and rectal cancer. The RHR cut-points that maximized the Youden index were 72.5 bpm for CRC, 72.5 bpm for colon, and 68.5 bpm for rectal cancer. All statistical tests were two-sided, with statistical significance determined by *p* < 0.05.

## 3. Results

### 3.1. Clinical Characteristics of Non-PS-Matched and PS-Matched CRC Patient Cohorts

Table 1 shows the clinical characteristics of the controls (*n* = 5909) and CRC patients (total *n* = 1241 with 854 colon cancer patients and 387 rectal cancer patients) included in the final analysis. The mean ages of the control and CRC patients were 47.3 ± 15.5 years and 62.6 ± 11.8 years, respectively. CRC, colon, and rectal cancer patients had significantly higher SBP, DBP, and glucose, and a significantly higher prevalence of HTN, diabetes, and dyslipidemia than controls (all *p* < 0.001). CRC and colon cancer patients had significantly higher white blood cell (WBC) counts (*p* < 0.001); RHR was significantly higher in CRC, colon, and rectal cancer patients than controls (72.7 bpm in CRC, 72.8 bpm in colon cancer, 72.3 bpm in rectal cancer, and 68.7 bpm in controls; all *p* < 0.001). A higher proportion of CRC and colon cancer patients were underweight/obese compared to controls. Proportion of current smokers and drinkers was higher among controls than among CRC patients.

Table 2 shows the clinical characteristics of CRC patients and controls after 1:1 propensity score (PS) matching for age and gender. A total of 1207 CRC patients (829 with colon cancer, 378 with rectal cancer) were matched with controls. After PS matching, CRC and colon cancer patients had higher SBP, DBP, glucose, and WBC than controls; however, there were no significant differences between these variables in rectal cancer patients and controls. RHR was significantly higher in CRC, colon, and rectal cancer patients than in controls. Weight status, presence of HTN, dyslipidemia, and alcohol/smoking status showed trends similar to those observed in non-PS-matched cases.

### 3.2. Associations between RHR and CRC

Figure 1 represents the mean ± SE of RHR in controls and CRC, colon, and rectal cancer patients in PS-matched cases after adjusting for age, gender, weight status, alcohol, smoking, HTN, diabetes, and dyslipidemia. RHR in CRC, colon, and rectal cancer were significantly higher than RHR in controls.

Table 3 shows the results of logistic regression analysis for CRC, colon, and rectal cancer according to RHR after adjusting for age, gender, weight status, alcohol, smoking, hypertension, diabetes, and dyslipidemia. In non-PS-matched cases, the ORs and 95% CIs for CRC, colon, and rectal cancer patients per RHR increment were 1.043 (1.036–1.049), 1.045 (1.037–1.053), and 1.040 (1.030–1.051), respectively, in unadjusted models and 1.043 (1.034–1.051), 1.046 (1.037–1.055), and 1.040 (1.027–1.052), respectively, in multivariable adjusted models. Among PS-matched cases, the ORs and 95% CIs for CRC, colon, and rectal cancer patients per RHR increment were 1.051 (1.042–1.060), 1.050 (1.039–1.061), and 1.053 (1.036–1.070), respectively, in unadjusted models and 1.044 (1.032–1.057), 1.046 (1.032–1.061), and 1.035 (1.008–1.063), respectively, in multivariable adjusted models.

We performed ROC analysis to evaluate whether RHR can predict CRC and determine optimal cut-point (Appendix A). The optimal RHR cut-points for predicting CRC, colon, and rectal cancer were 72.5, 72.5, and 68.5 bpm, respectively.

Figure 2A,B show the prevalence of above-cut-point RHR among CRC, colon, and rectal cancer patients and among controls. Among non-PS-matched cases, the prevalence of a RHR ≥ 72.5 bpm, ≥72.5 bpm, ≥68.5 bpm was significantly higher among CRC, colon, and rectal cancer patients than among controls (48.9% vs. 24.6%, *p* < 0.001; 50.4% vs. 24.5%, *p* < 0.001; and 61.2% vs. 39.0%, *p* < 0.001, respectively) (Figure 2A). Among PS-matched cases, the prevalence of a RHR ≥ 72.5 bpm, ≥72.5 bpm, ≥68.5 bpm, was significantly higher among CRC, colon, and rectal cancer patients than among controls (49.1% vs. 21.0%, *p* < 0.001; 50.7% vs. 21.8%, *p* < 0.001; and 61.1% vs. 31.2%, *p* < 0.001, respectively) (Figure 2B).

## 4. Discussion

In this case-controlled study, we found that patients with CRC, colon, and rectal cancer have a significantly higher RHR compared to cancer-free controls among both non-PS-matched cases and cases PS-matched for age and gender. Elevated RHR was independently associated with the presence of CRC, colon, and rectal cancer. The prevalence of CRC, colon, and rectal cancer were significantly higher in cases with RHR ≥ 72.5 bpm, ≥72.5 bpm, and ≥68.5 bpm, respectively, than in controls.

Sympathetic nervous system (SNS) activation is regulated by β-adrenergic signaling [17], which influences tumor microenvironment and molecular pathways that contribute to cancer development and progression [17,30]. These lines of evidence prompted the use of beta-blockers, which block the action of catecholamines and norepinephrine, for preventing/treating cancer. Epidemiologic studies have shown that higher RHR is associated with cancer-related—and all-cause—mortality in cancer patients [16,21]. A recent meta-analysis revealed that beta-blocker use after diagnosis has a positive impact on cancer patients’ overall survival [31]. Despite these studies supporting an association between RHR and cancer-related mortality/prognosis, no study has conclusively demonstrated a relationship between beta-blocker use and CRC-specific mortality. A prospective study conducted in Netherlands did not show a significant association between beta-blocker use before and after diagnosis and CRC prognosis [22]. A nested case-controlled study involving the UK Clinical Practice Research Datalink cohort also failed to demonstrate an association between beta-blocker use after diagnosis and CRC-specific mortality [23]. By contrast, some studies, such as van Kruijsdijk et al. [20] showed the possibility of a link between RHR and CRC. A study conducted in Germany also demonstrated an association between RHR and prognosis of advanced CRC patients [21]. Interestingly, a Korean study found that CRC survivors with higher RHR had an independently elevated risk of advanced adenoma recurrence than those with lower RHR [25].

Several potential mechanisms could explain our observed association between RHR and CRC prevalence. Insulin and insulin-like growth factor play important roles in the pathogenesis and progression of CRC [32]. Sympathetic tone overactivity causes insulin resistance through both beta-receptors and alpha-receptors [10]. Infusion of a β-adrenergic agonist affects muscle fiber type in rats and leads to an increase of insulin-resistant fast-twitch muscles [33]. Alpha-adrenergic stimulation causes a vasoconstriction-mediated decrease in glucose utilization by human skeletal muscles [34]. Insulin and insulin resistance pathways are crucial mediators of the pathophysiology that lead to malignant transition of normal colorectal epithelial cells to adenoma or adenocarcinoma [32,35]. In the present study, CRC and colon cancer patients had a higher level of glucose and higher prevalence of diabetes than controls; however, there were no significant differences in these variables between rectal cancer patients and controls. We found a relatively weaker association between RHR and rectal cancer (OR = 1.035; 95% CI: 1.008–1.063) compared to the association between RHR and colon cancer after PS matching. The colon and rectum differ in location and innervation and serve different functions [36]. Furthermore, many studies have established that obesity, insufficient physical activity, and red meat consumption are important risk factors for colon cancer but not for rectal cancer [37,38]. The nurses’ health study (NHS) and health professionals follow-up study (HPFS) have suggested that colon cells have a different susceptibility for insulin growth factors and different insulin sensitivity compared to rectal cells [37]. Given that any divergences in the mechanisms underlying development of colon and rectal cancer are currently unclear, further studies are needed to confirm the effect of RHR on CRC considering the differences between colon and rectal cancer.

In our study, adjustment for potential confounders resulted only in a minor change in the value of the OR compared to that obtained with unadjusted models; this suggests that RHR itself is still likely an important factor in CRC development although we could not consider all possible confounders. Our study used ROC analysis to determine the best cut-points to predict CRC, colon, and rectal cancer. The areas under the curve (AUCs) were 0.639 (95% CI: 0.617–0.661) with cut-point at 72.5 bpm for CRC, 0.638 (95% CI: 0.611–0.665) at 72.5 bpm for colon cancer, and 0.641 (95% CI: 0.601–0.681) at 68.5 bpm for rectal cancer. These results are similar to those from a previous study that found that an RHR of 72 bpm had the highest sensitivity and specificity with AUC = 0.593 (95% CI: 0.509–0.764) for survival in cancer patients [21]. However, since the value of AUC was not high enough to predict CRC in our results, we could not use RHR value as a screening method to predict CRC. Further longitudinal prospective studies are needed to confirm the predictive value of RHR for CRC. Our results should be interpreted with caution because of our study’s cross-sectional nature, which cannot reveal causality. Therefore, we could not determine whether the elevated RHR in CRC is just an epiphenomenon arising from cancer-related worsening of the CRC patients’ general condition. Furthermore, our results have limited generalizability as our data were collected from a single hospital in South Korea. Third, we had limited information on the use of antihypertensive medications such as beta-blockers. Although this is may be a potential confounder, the proportion of patients with HTN (who are likely to be prescribed these medications) was significantly higher among CRC patients than among controls. Moreover, patients with angina pectoris or myocardial infarction (who are likely to use drugs that could affect their RHR) were excluded from our study. Therefore, it is unlikely that the lack of information regarding medications has had a crucial impact on the results. Fourth, RHR was measured by physician, not by echocardiogram. In addition, RHR in KNHANES calculated by counting the pulse for 15 s and then multiplying it by 4. Kobayashi et al. [39] showed that pulse-counting in shorter time periods introduces the possibility of an error within clinically acceptable range. RHR by pulse pulsation in shorter time is a routinely used method of measurement in clinical settings. Fifth, we only investigated the association between RHR and confirmed CRC. The relationship between RHR and colorectal polyps or any precancerous diseases could also be important issues. Strengths of our study include the following: (a) We found differences in RHR between PS-matched CRC patients and cancer-free controls before surgery or any other treatment was administered to the patients, and (b) associations between RHR and CRC were evaluated after taking into consideration anatomical location of the tumor (colon vs. rectum).

## 5. Conclusions

RHR is significantly higher in CRC, colon, and rectal cancer patients compared to disease-free controls. RHR is independently associated with presence of CRC, colon, and rectal cancer. Further studies are needed to clarify if RHR plays any causal role in CRC development, and whether decreasing RHR may be an effective strategy to reduce CRC risk.

## Figures and Tables

**Figure 1 ijerph-16-02883-f001:**
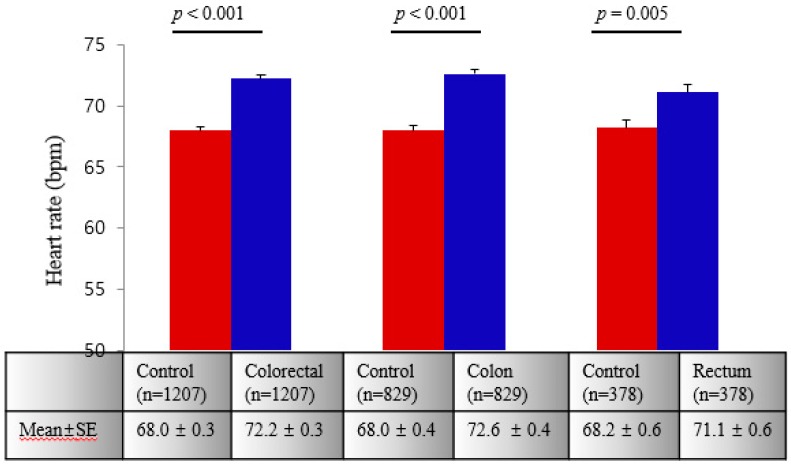
The mean ± standard error (SE) of resting heart rate in CRC, colon, and rectal cancer compared with controls. *p* was calculated using a linear mixed model after adjusting for age, sex, weight, alcohol consumption, smoking history, hypertension, diabetes, and dyslipidemia.

**Figure 2 ijerph-16-02883-f002:**
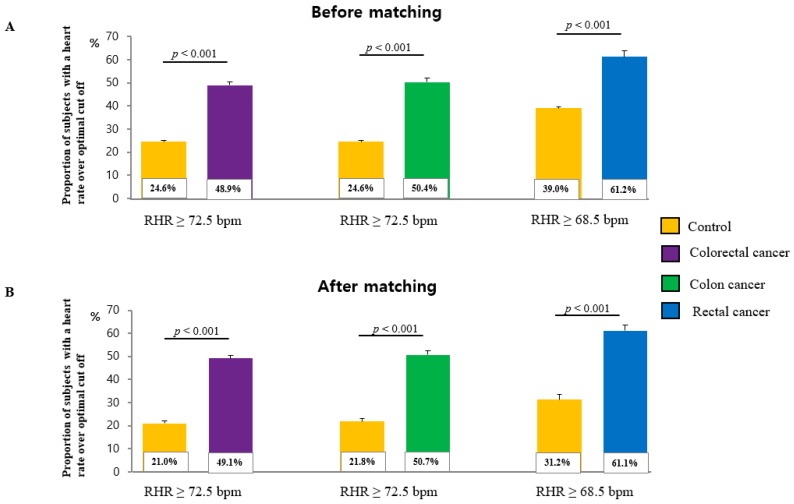
Prevalence of CRC, colon, and rectal cancer across each cut-off points (**A**) before propensity score matching and (**B**) after propensity score matching.

**Table 1 ijerph-16-02883-t001:** Clinical characteristics of colorectal cancer (CRC) patients and controls.

Variable	Control	Cases			*p*		
		Colorectal	Colon	Rectum	^†^P	^‡^P	^§^P
N	5909	1241	854	387			
Age (years)	47.3 ± 15.5	62.6 ± 11.8	63.4 ± 11.5	61.0 ± 12.4	*	*	*
Gender					*	*	*
Male	2836 (48.0)	712 (57.4)	464 (54.3)	248 (64.1)			
Female	3073 (52.0)	529 (42.6)	390 (45.7)	139 (35.9)			
SBP (mmHg)	116.5 ± 15.6	126.0 ± 15.1	126.2 ± 15.1	125.4 ± 15.0	*	*	*
DBP (mmHg)	75.2 ± 10.2	77.2 ± 10.3	77.1 ± 10.3	77.5 ± 10.2	*	*	*
RHR (bpm)	68.7 ± 9.0	72.7 ± 11.3	72.8 ± 11.3	72.3 ± 11.3	*	*	*
Weight status					*	*	NS
Underweight	2386 (40.4)	555 (44.7)	380 (44.5)	175 (45.2)			
Normal	1442 (24.4)	308 (24.8)	217 (25.4)	91 (23.5)			
Overweight	1844 (31.2)	310 (25.0)	210 (24.6)	100 (25.8)			
Obese	237 (4.0)	68 (5.5)	47 (5.5)	21 (5.4)			
WBC (103 cells)	6.2 ± 1.3	6.4 ± 1.5	6.6 ± 1.5	6.1 ± 1.4	*	*	NS
Glucose (mg/dL)	98.9 ± 21.1	109.4 ± 31.2	110.3 ± 32.5	107.5 ± 28.2	*	*	*
TC (mg/dL)	188.1 ± 33.7	176.1 ± 36.7	173.3 ± 35.7	182.1 ± 38.2	*	*	*
HTN, n (%)	913 (15.4)	518 (41.7)	374 (43.8)	144 (37.2)	*	*	*
DM, n (%)	371 (6.3)	225 (18.1)	161 (18.8)	64 (16.5)	*	*	*
Dyslipidemia, n (%)	405 (6.8)	200 (16.1)	147 (17.2)	53 (13.7)	*	*	*
Alcohol					*	*	*
Never	602 (10.2)	688 (55.4)	492 (57.6)	196 (50.7)			
Ex	804 (13.6)	330 (26.6)	187 (21.9)	143 (37.0)			
Current	4503 (76.2)	223 (18.0)	175 (20.5)	48 (12.4)			
Smoking					*	*	*
Never	3383 (57.3)	764 (61.5)	552 (64.6)	212 (54.8)			
Ex	1183 (20.0)	362 (29.2)	222 (26.0)	140 (36.2)			
Current	1343 (22.7)	115 (9.3)	80 (9.4)	35 (9.0)			
TNM stage							
0	-	326 (26.3)	187 (21.9)	139 (35.9)			
I	-	365 (29.4)	270 (31.6)	95 (24.6)			
II	-	359 (28.9)	257 (30.1)	102 (26.4)			
III	-	191 (15.4)	140 (16.4)	51 (13.2)			

SBP, systolic blood pressure; DBP, diastolic blood pressure, WBC, white blood cell; TC, total cholesterol; HTN, hypertension; DM, diabetes; Bpm, beat per minute. Weight status is categorized as follows; underweight (<18.5 kg/m^2^), normal (18.5–23 kg/m^2^), overweight (23–25 kg/m^2^), and obese (>25 kg/m^2^). Data are presented as mean ± standard deviations (SDs) or number (%). *p* is calculated as independent t-test for continuous variables and chi-square test for categorical variables. ^†^P; control vs. colorectal, ^‡^P: Control vs. colon, ^§^P: Control vs. rectum. Asterisk (*) means *p* is less than 0.01. NS; non-significance.

**Table 2 ijerph-16-02883-t002:** Clinical characteristics of colorectal cancer patients and controls after 1:1 propensity score matching for age and gender.

Variable	Control	Colorectal	*p*	Control	Colon	*p*	Control	Rectum	*p*
N	1207	1207		829	829		378	378	
Age (years)	62.0 ± 11.4	62.0 ± 11.4	NS	62.7 ± 11.1	62.7 ± 11.1	NS	60.4 ± 12.0	60.4 ± 12.0	NS
Gender			NS			NS			NS
Male	695 (57.6)	695 (57.6)		452 (54.5)	452 (54.5)		243 (64.3)	243 (64.3)	
Female	512 (42.4)	512 (42.4)		377 (45.5)	377 (45.5)		135 (35.7)	135 (35.7)	
SBP (mmHg)	123.6 ± 16.1	125.6 ± 15.0	*	123.4 ± 16.3	126.1 ± 15.0	*	124.2 ± 15.9	125.4 ± 15.0	NS
DBP (mmHg)	75.6 ± 10.3	77.3 ± 10.3	*	75.4 ± 10.2	77.2 ± 10.3	*	76.3 ± 10.4	77.5 ± 10.2	NS
RHR (bpm)	67.6 ± 9.2	72.7 ± 11.3	*	67.8 ± 9.2	72.9 ± 11.4	*	67.1 ± 9.1	72.2 ± 11.3	*
Weight status			*			*			*
Underweight	28 (2.3)	64 (5.3)		18 (2.2)	45 (5.4)		10 (2.6)	19 (5.0)	
Normal	423 (35.1)	543 (45.0)		302 (36.4)	372 (44.9)		121 (32.0)	171 (45.2)	
Overweight	338 (28.0)	302 (25.0)		226 (27.3)	211 (25.5)		112 (29.6)	91 (24.1)	
Obese	418 (34.3)	298 (24.7)		283 (34.1)	201 (24.3)		135 (35.7)	97 (25.7)	
WBC (103 cells)	6.2 ± 1.3	6.4 ± 1.5	*	6.1 ± 1.3	6.6 ± 1.5	*	6.2 ± 1.3	6.1 ± 1.4	NS
Glucose (mg/dL)	103.6 ± 22.0	109.3 ± 31.1	*	102.9 ± 20.7	110.2 ± 32.3	*	105.2 ± 24.6	107.5 ± 28.1	NS
TC (mg/dL)	191.2 ± 34.7	177.0 ± 36.4	*	192.0 ± 34.7	174.4 ± 35.3	*	189.5 ± 34.6	182.6 ± 38.2	*
HTN, n (%)	381 (31.6)	497 (41.18)	*	278 (33.5)	357 (43.1)	*	103 (27.2)	140 (37.0)	*
DM, n (%)	148 (12.3)	216 (17.90)	*	102 (12.3)	157 (18.9)	*	46 (12.2)	59 (15.6)	NS
Dyslipidemia,n (%)	130 (10.77)	193 (16.0)	*	100 (12.1)	141 (17.0)	*	30 (7.9)	52 (13.8)	*
Alcohol			*			*			*
Never	189 (15.7)	659 (54.6)		149 (18.0)	470 (56.7)		40 (10.6)	189 (50.0)	
Ex	220 (18.2)	325 (26.9)		151 (18.2)	184 (22.2)		69 (18.2)	141 (37.3)	
Current	798 (66.1)	223 (18.5)		529 (63.8)	175 (21.1)		269 (71.2)	48 (12.7)	
Smoking			*			*			*
Never	591 (49.0)	740 (61.3)		430 (51.9)	533 (64.3)		161 (42.6)	207 (54.8)	
Ex	351 (29.1)	352 (29.2)		237 (28.6)	216 (26.1)		114 (30.2)	136 (36.0)	
Current	265 (21.9)	115 (9.5)		162 (19.5)	80 (9.6)		103 (27.3)	35 (9.2)	
TNM stage									
0	-	320 (26.5)		-	184 (22.2)		-	136 (36.0)	
I	-	353 (29.3)		-	260 (31.4)		-	93 (24.6)	
II	-	345 (28.6)		-	246 (29.7)		-	99 (26.2)	
III	-	189 (15.7)		-	139 (16.8)		-	50 (13.2)	

Data are presented as mean ± standard deviations (SDs) or number (%). SBP, systolic blood pressure; DBP, diastolic blood pressure; WBC, white blood cell; TC, total cholesterol; HTN, hypertension; DM, diabetes; Bpm, beat per minute. Weight status is categorized as follows; underweight (<18.5 kg/m^2^), normal (18.5–23 kg/m^2^), overweight (23–25 kg/m^2^), and obese (>25 kg/m^2^). * *p* is calculated via paired t-test for continuous variables and McNemar’s test for categorical variables. Asterisk (*) means *p* is less than 0.01. NS; non-significance.

**Table 3 ijerph-16-02883-t003:** Logistic regression models for CRC according to resting heart rate (RHR) in before and after PS matching.

Model	Colorectal Cancer		Colon Cancer		Rectal Cancer	
	OR (95% CIs)	*p*	OR (95% CIs)	*p*	OR (95% CIs)	*p*
Before PS matching						
Univariable *	1.043 (1.036–1.049)	<0.001	1.045 (1.037–1.053)	<0.001	1.040 (1.030–1.051)	<0.001
Multivariable ^†^	1.043 (1.034–1.051)	<0.001	1.046 (1.037–1.055)	<0.001	1.040 (1.027–1.052)	<0.001
After PS matching						
Univariable **	1.051 (1.042–1.060)	<0.001	1.050 (1.039–1.061)	<0.001	1.053 (1.036–1.070)	<0.001
Multivariable ^‡^	1.044 (1.032–1.057)	<0.001	1.046 (1.032–1.061)	<0.001	1.035 (1.008–1.063)	<0.001

* OR and 95% CIs were calculated using simple logistic regression analysis. ** OR and 95% CIs were calculated using conditional simple logistic regression analysis. ^†^ OR and 95% CIs were calculated using multiple logistic regression analysis after adjusting for age, gender, weight status, smoking, alcohol, hypertension, diabetes, and dyslipidemia. ^‡^ OR and 95% CIs were calculated using conditional multiple logistic regression analysis after adjusting for age, gender, weight status, smoking, alcohol, hypertension, diabetes, and dyslipidemia.

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
