# Peer review of "Association between Resting Heart Rate and Colorectal Cancer: Results from a Case-Controlled Study"

_ijerph, 2019, doi:10.3390/ijerph16162883_

Round 1

Reviewer 1 Report

 The data reported in the manuscript: Association between resting heart rate and colorectal 3 cancer: Results from a case-controlled study from-Jin Kwon et al., as very nicely written, scientifically sound, data presentation are exhaustively explained, thus recommending its publication on IJERPH Journal without any revisions. I will ask the authors only a question about the research design, I'm interested to know if they indagate also the correlation of resting heart rate, with patient affected form polyposis or different precancerous disease, such as the different grade of dysplasia.

Author Response

 Reviewer #1

The data reported in the manuscript: Association between resting heart rate and colorectal 3 cancer: Results from a case-controlled study from-Jin Kwon et al., as very nicely written, scientifically sound, data presentation are exhaustively explained, thus recommending its publication on IJERPH Journal without any revisions.

I will ask the authors only a question about the research design, I'm interested to know if they indagate also the correlation of resting heart rate, with patient affected form polyposis or different precancerous disease, such as the different grade of dysplasia.

Response: We fully agree with you in that correlations between resting heart rate and polyposis or other precancerous diseases are also important. A previous Korean study has demonstrated the association between resting heart rate and recurrent colorectal polyp [1]. In the current study, we only included patients who were confirmed to have colorectal cancer. Upon your understanding, we would like to state this as a limitation in the discussion section as follows:

Fifth, we only investigated the association between RHR and confirmed CRC. The relationship between RHR and colorectal polyps or any precancerous diseases could also be important issues.

Reference

Park, J.; Kim, J.H.; Park, Y.; Park, S.J.; Cheon, J.H.; Kim, W.H.; Park, J.S.; Jeon, J.Y.; Kim, T.I. Resting heart rate is an independent predictor of advanced colorectal adenoma recurrence. PloS one 2018, 13, e0193753.

Reviewer 2 Report

This study reports on differences in resting heart rate (RHR) between colorectal cancer (CRC) patients and health controls. This study adds to the growing literature on RHR and colon cancer risk and prognosis. The main concern for this manuscript is that the study motivation needs to be more clearly defined. Why does the difference in RHR between prevalent CRC and disease-free controls matter? The fact that this comparison has never been done before is not enough. I also have several minor points that require attention, listed below.

Lines 53-54 need references: “However, although studies have suggested links between high RHR and CRC risk, there are conflicting reports regarding the association between RHR and CRC.“

Line 84: “….and patients with a combination of colon and rectal cancer were defined as CRC patients.” should be rephrased as “….the total cohort of patients with colon or rectal cancer were defined as CRC patients”. 

Is there a reference for the validity of comparing RHR measurements based on two different time intervals (e.g., 15 seconds x 4 vs. 60 seconds)?

What do the authors mean on lines 91-92: “RHR was measured for 60 seconds when the RHR was irregular (e.g. less than 15 times for 15 seconds or more than 26 times for 15 seconds).”

What was the random effect in the linear mixed model?

The value added for analyzing colon and rectal cancer separately is unclear. The authors should consider moving these analyses to supplemental materials, as they make the tables very busy.

It is redundant to include both n (%) values for binary covariates. Removing the unnecessary rows will make the tables cleaner.

Did the authors consider any potential interaction effects between RHR and other study covariates?

What is the RHR increment that corresponds to the ORs in Table 3?

The difference in mean RHR between cases and controls was about 4 bpm - is this clinically meaningful? Although it is statistically significant, if it is not clinically significant then the interpretation of the results should be revised accordingly.

What was the rationale for determining a RHR cut-point for CRC prediction? Based on the ROC figures, it does not appear that RHR is able to distinguish between cases and controls very well. What was the sensitivity and specificity of the specified threshold? 

Line 264 should read “RHR is significantly higher in CRC, colon, and rectal cancer patients compared with disease-free controls”.

Reviewer 3 Report

In this paper Kwan et al demonstrated that patients with colorectal cancer (CRC), irrespectively from its location, have higher resting heart rates (RHR). The analysis was performed after propensity score matching, therefore it is presumable that patients bias selection was avoided. Main comments:

1) Page 2 line 76: were arrhythmia patients excluded?

2) Page 3 lines 95-98: why Authors adopted this cut-off for BMI? Usually, obesity is defined as BMI>30, and not >25 as herein reported.

3) The fact that the difference in RHR between controls and cases is low (about 5 bpm) may explain the fact that AUCs are extremely disappointing; an AUC<0.75 is too low, therefore RHR cannot be considered as an useful method to screen CRC. Please discuss.

4) Additionally, OR values, despite being significant, are very close to 1, therefore the incremental risk of CRC in relation to RHR is very low. Therefore it is hypothesizable that the positive relationship may be explained by the fact that metabolic syndrome and diabetes, which are well known risk factors for CRC, are usually associated to high RHR. Therefore, the link between RHR and CRC is indirect. Please discuss.

Author Response

Reviewer #3

In this paper Kwan et al demonstrated that patients with colorectal cancer (CRC), irrespectively from its location, have higher resting heart rates (RHR). The analysis was performed after propensity score matching, therefore it is presumable that patients bias selection was avoided. Main comments:

1) Page 2 line 76: were arrhythmia patients excluded?

Response: We apologize for not describing the exclusion criteria in more detail. We excluded the patients with diagnosis of arrhythmia by physician, RHR>100, and RHR<40.

We corrected the exclusion criteria as follows:

Patients with the following conditions were also excluded: (a) a history of malignancy, (b) CVD (e.g., stroke, angina, myocardial infarction), (c) other diseases that can influence RHR (e.g., thyroid diseases), (d) body mass index (BMI) <15 kg/m2, (e) arrhythmia history, RHR >100 bpm, and <40 bpm, and (f) systolic blood pressure (SBP) ≥180 mmHg or diastolic blood pressure (DBP) ≥120 mmHg.

2) Page 3 lines 95-98: why Authors adopted this cut-off for BMI? Usually, obesity is defined as BMI>30, and not >25 as herein reported.

Response: We defined obesity based on the BMI criteria for Asians of the Regional Office for the Western Pacific Region of World Health Organization (underweight (<18.5 kg/m2), normal (18.5–23 kg/m2), overweight (23–25 kg/m2), and obese (>25 kg/m2)) [13]. Asian population has been known to have a greater risk of metabolic diseases compared to Western population, even at similar BMI level [14]. [14]. In addition, we found no differences in the results when we set the obesity as BMI>30.

3) The fact that the difference in RHR between controls and cases is low (about 5 bpm) may explain the fact that AUCs are extremely disappointing; an AUC<0.75 is too low, therefore RHR cannot be considered as an useful method to screen CRC. Please discuss.

Response: We fully understand your concerns. We also agree that the cut-off point of RHR, presented in the current study, could not be useful for screening CRC.

We first intended to check the RHR point at which the risk of CRC rises, and to find CRC prevalence based on that point. Therefore, we inserted ROC curve as a supplementary file.

We would like to add the following details to the discussion section:

However, since the value of AUC was not high enough to predict CRC in our results, we could not use RHR value as a screening method to predict CRC. Further longitudinal prospective studies are needed to confirm the predictive value of RHR for CRC.

4) Additionally, OR values, despite being significant, are very close to 1, therefore the incremental risk of CRC in relation to RHR is very low. Therefore it is hypothesizable that the positive relationship may be explained by the fact that metabolic syndrome and diabetes, which are well known risk factors for CRC, are usually associated to high RHR. Therefore, the link between RHR and CRC is indirect. Please discuss.

Response: We agree that the positive relationship between CRC and RHR can be explained by metabolic syndrome, diabetes, and other metabolic diseases. However, we found an independent and significant association between CRC and RHR after adjusting for weight status, hypertension, diabetes, and dyslipidemia. In our study, adjustment for potential confounders resulted in only a minor change in the value of OR compared to that obtained in unadjusted models. This suggests that RHR itself is still likely an important factor in CRC development, although we could not consider all possible confounders.

In our study, adjustment for potential confounders resulted in only a minor change in the value of OR compared to that obtained in unadjusted models. This suggests that RHR itself is still likely an important factor in CRC development, although we could not consider all possible confounders.

References

13. Organization, W.H. The asia-pacific perspective: Redefining obesity and its treatment. Sydney: Health Communications Australia: 2000.

14. Pan, W.H.; Yeh, W.T.; Weng, L.C. Epidemiology of metabolic syndrome in asia. Asia Pacific journal of clinical nutrition 2008, 17 Suppl 1, 37-42.

Round 2

Reviewer 2 Report

The authors address all concerns and I have no additional critques.

Reviewer 3 Report

All answers were satisfactory